# A Hierarchical Porous Cellulose Sponge Modified with Chlorogenic Acid as a Antibacterial Material for Water Disinfection

**En-Jiang Liu, Jia-Xing Huang, Run-Ze Hu, Xiao-Hui Yao, Wei-Guo Zhao, Dong-Yang Zhang and Tao Chen ***

College of Biotechnology and Sericultural Research Institute, Jiangsu University of Science and Technology, Zhenjiang 212018, China
* Correspondence: 199000001909@just.edu.cn; Tel.: +86-511-85616777

**Abstract:** Water contaminated by microorganisms will seriously endanger public safety, as many diseases are caused by microorganisms, and water disinfection materials offer an effective method to solve this problem. In this work, a hierarchical porous structure cellulose sponge (CS) was constructed as the water disinfection filter substrate, where "long−chain" cellulose served as the skeleton to construct major pores, and "short−chain" cellulose filled the gaps between "long−chain" cellulose to construct minor pores. After CS was covalently modified by chlorogenic acid (CGA) to fabricate cellulose–chlorogenic acid sponge (C−CGAS), a hierarchical porous structure was retained. Due to the hierarchical porous structure, C−CGAS showed good mechanical stability (2.84% unrecoverable strain after 1000 compression cycles). Furthermore, C−CGAS also showed good antibacterial and antifungal abilities due to the antimicrobial ability and high water flux, and C−CGAS could eliminate 95% of *E. coli* within 0.5 h in the water disinfection test. Due to the stable covalent modification of CGA and its mechanical stability, C−CGAS showed no breakage, and even after nine consecutive use cycles, the antibacterial properties were almost unchanged. Thus, C−CGAS is a reusable and highly efficient water disinfection material. This study provides a new approach for the preparation of recyclable, safe, and efficient water disinfection materials.

**Keywords:** cellulose; hierarchical porous cellulose sponge; chemical modification; chlorogenic acid; mechanical stability; water disinfection

## 1. Introduction

Safe and clean drinking water is a basic need for human survival, and water quality significantly affects human health [1–3]. However, according to a World Health Organization (WHO) report, approximately 30% of people worldwide lack clean water sources, and more than 2 million people die each year due to water contamination [4]. Water contamination by pathogenic microbes is a severe imperilment to the security of global public health [5–7]; therefore, an effective and safe method to disinfect water should be urgently developed.

Many methods have been recently developed to disinfect water, and the direct addition of disinfectants to water has been shown to be a common and effective method; however, disinfectants are difficult to reuse [8]. Furthermore, the production of disinfection by−products and the excessive use of disinfectants can have a serious impact on human health [9]. To solve this problem, many studies have focused on loading antibacterial substances onto substrate materials, among which polymer membrane−based filtration has been widely used [10–12], and antibacterial agents can be integrated into the polymer membrane matrix to exert antibacterial effects. However, biofouling on membrane surfaces has become a potential pathogenic contaminant risk, and the low−porosity structures of the membrane will significantly decrease water flux as well as the water disinfection efficiency [13]. By increasing the porosity, water flux can be increased, and bacterial stasis can be effectively avoided; however, the increase in porosity has often been accompanied

by a decrease in mechanical strength. Therefore, filter substrate materials need to maintain mechanical properties while maintaining high water flux. In addition, the safety and stable presence of antimicrobial agents in the substrate are important for recycling use. Currently, most water purification materials have been developed with cationic polymers [14], antibacterial peptides [15], and metal nanoparticles [16] as antibacterial agents [14]. Although these antibacterial agents have shown excellent antibacterial effects, their toxicity and unstable immobilization are still the main issues that limit the further application of water purification materials [17]. Thus, it is necessary to develop a novel water disinfection material with high water flux and adequate mechanical stability. Furthermore, the antibacterial agents must be biosafe and stably connected to the substrate materials.

Cellulose is a semi−crystalline biopolymer of cellobiose, composed of two anhydroglucose units ($C_6H_{10}O_5$) joined by a β−glycosidic bond [18–21]. Due to its environmental friendliness, renewability, safety, and mechanical properties, cellulose has been fabricated into cellulose membranes, cellulose hydrogels, and cellulose sponges (CS) [22–25], as well as widely used in food packaging, textile biodegradability, and other fields [26,27]. Among these, CS have shown potential as water disinfection substrates due to their high porosity, hydrophilicity, and mechanical strength [28]. Based on the above characteristics, CS−based filters combine a large water flux and water flow rate, which requires that the antibacterial agent be immobilized stably to ensure a good antibacterial effect for a "long time". Cellulose contains a large number of hydroxyl groups on its surface, offering stable modification sites for functional groups with antibacterial activity [1]. Moreover, many studies have used epichlorohydrin (EPI) to crosslink hydroxyl groups on cellulose surfaces to achieve self−crosslinking of cellulose [29–31] or introduce other functional compounds, such as fluorinated polymers [32], clay nanocomposites [33], linseed gum [34], etc., which improve the adsorption, antibacterial, and mechanical properties of cellulose materials. In addition, cellulose exhibits hydrophilicity, making it conducive to the diffusion of water. This allows antibacterial groups to effectively contact microorganisms in water, improving the water disinfection performance of cellulose. Therefore, cellulose may be an ideal water disinfection substrate material. However, the specific surface area (SSA) of common cellulose, such as microcrystalline cellulose (MCC), is low and antibacterial agents are difficult to immobilize. Thus, in this study, MCC was dissolved in NaOH/urea aqueous solution and restructured a high SSA cellulose sponge with nanopores. Moreover, to enhance the mechanical properties, absorbent cotton (AC) was added to construct major pores by microscale fibers, and the hierarchical porous structure cellulose sponge (CS) was prepared. The structure of hierarchical porous could provide adequate elasticity and mechanical properties, but many studies have used at least two different materials [35–37]. In this study, a hierarchical porous structure cellulose sponge was constructed based on cellulose with different molecular weights, and adequate elastic and mechanical stability were obtained.

Chlorogenic acid (CGA) is 3−O−caffeoylquinic acid, which is a natural component in mulberry (*Morus alba* L.), honeysuckle (*Lonicerae Japonicae* Thunb.), and *Eucommia ulmoides* leaves (*Eucommia ulmoides* Oliv.) [38]. CGA exhibits adequate antimicrobial activity on bacteria and fungi [39–42], and has been widely used in health products and food additives, suggesting sufficient biological safety [40,43,44]. Moreover, CGA can be introduced into a variety of polysaccharides substrates, such as chitin [45], starch [46], cellulose, etc. [47]; thus, CGA is an ideal antimicrobial agent. As a result, CS stably modified with CGA as water disinfection materials can combine high water flux, recycling stability, antibacterial ability, and biosafety.

In this study, CS covalently modified with CGA were successfully prepared for use as water disinfection materials. Based on the porous structure of CS and the antibacterial properties of CGA, the filter could effectively sterilize in a short time. In addition, due to the stable modification of CGA and the mechanical strength of the CS, the filter still had sufficient water disinfection ability after multiple recycling iterations. Benefiting from favorable antibacterial and mechanical properties, high water flux, long service life,

and non−toxic properties, this CS may provide new ideas for the preparation of water disinfection materials.

## 2. Materials and Methods

### 2.1. Materials and Reagents

Anhydrous sodium sulfate ($Na_2SO_4$), anhydrous sodium carbonate ($Na_2CO_3$), epichlorohydrin (EPI), sodium hydroxide (NaOH), urea, and absolute ethanol ($C_2H_5OH$) were purchased from Sinopharm Chemical Reagent Co., Ltd., Shanghai, China. Absorbent cotton (AC) was purchased from Qingdao Shuertz Biotechnology Co., Ltd., Qingdao, China. MCC (particle size 25 μm) was purchased from Sinopharm Chemical Reagent Co., Ltd., Shanghai, China. In addition, chlorogenic acid (CGA) (Q, 95% pure) was produced by Shanghai Aladdin Biochemical Technology Co., Ltd., Shanghai, China, and deionized water was produced by a Milli−Q purification system (Millipore, Burlington, MA, USA). The chemicals and solvents were of analytical grade and used without further purification.

*E. coli* (ATCC 25922, $G^−$) and *S. aureus* (ATCC 6538, $G^+$) were produced by Shanghai Luwei Technology Co., Ltd., Shanghai, China. *Rhizopus stolonifera* (CGMCC 3.31) was purchased from Beijing Microbiological Culture Collection Center, China. LB broth powder (AL1011) was produced by Shanghai Acmec Biochemical Co. Ltd., Shanghai, China. In addition, phosphate−buffered saline (PBS) was purchased from Sangon Biotech Co., Ltd., Shanghai, China, and agar was purchased from Beijing BioDee Biotechnology Co. Ltd., Beijing, China.

### 2.2. Characterization

A field emission scanning electron microscope (QUANTA 250 FEG, Thermo Fisher, Waltham, MA, USA) was used to observe the morphological structures of the sponges. Fourier transform infrared spectroscopy (FTIR) was used to determine the chemical cellulose–chlorogenic acid sponge (C−CGAS) structures (Nicolet iS5, Thermo Fisher, Waltham, MA, USA), and the surface chemical compositions of CS and C−CGAS were investigated by X−ray photoelectron spectroscopy (XPS) (AXIS, Shimadzu, Kyoto, Japan). The dynamic water contact angles (WCAs, 4 μL) of CS and C−CGAS were determined by a contact angle goniometer (Kino SL 250). The mechanical properties of CS and C−CGAS were determined through dynamic compressive testing by dynamic mechanical analysis (DMA) (QLW−5E, Xiamen Qunlong Scientific Instrument Co., Ltd., Xiamen, China). The test instrument was equipped with a parallel−plate compression clamp, where the load cell was set to 50 N, and the compression speed was 12 mm/min. All samples were tested under controlled conditions (25 °C and 60% RH).

### 2.3. Preparation of the CS

First, 12 wt % urea and 7 wt % sodium hydroxide aqueous solutions were prepared and pre−frozen at −12 °C. Then, 3 g of AC and 3 g of MCC were dissolved in 97 g of the precooled solution and stirred to fully dissolve. Then, the AC and MCC solutions were mixed at a weight ratio of 1:1 with sufficient stirring to completely mix the solution. Subsequently, 80 wt % $Na_2SO_4$ was added to the mixed solution, thoroughly stirred, then filled in the mold and frozen at −70 °C for 12 h followed by soaking in 80 °C deionized water for 48 h to obtain CS.

### 2.4. Preparation of the C−CGAS

The CS was mixed with 50 mL of EPI and 200 mL of NaOH solution at a concentration of 1 mol/L, and then the mixture was placed in a 25 °C water bath and stirred for 12 h. Afterward, the CS was completely washed with deionized water and immersed in 100 mL of 2% $Na_2CO_3$ solution with added 10 mL of 0.04 g/mL ethanolic CGA solution. Afterward, the mixture reacted in a shaking incubator at 150 r/min and 50 °C for 12 h. Finally, C−CGAS was washed completely with deionized water. Scheme 1 shows the modification of the CS by CGA.

**Scheme 1.** Modification of the CS by CGA and the (**a–d**) four possible structures of C−CGAS.

### 2.5. Preparation of the Culture Medium and Bacterial Activation

All supplies were sterilized in an autoclave in advance. The CS and C−CGAS were sterilized by UV light for 2 h. Then, 35 g of LB broth powder and 950 mL of sterile water were mixed, and the pH of the medium was adjusted to 7.4 by using 1 mol/L NaOH solution, followed by dilution with sterile water to a final volume of 1 L. For the solid LB medium, 15 g of agar was added, and the other steps were the same as mentioned above. Then, the liquid and solid media were sterilized at 121 °C for 20 min. After culturing the bacteria in the LB medium at 37 °C and 200 r for 24 h, the suspension was diluted with fresh LB medium at a volume ratio of 1:100, and the mixture was cultured in an incubator shaker at 37 °C and 200 r. The optical density was determined at 600 nm (OD600) every 2 h, and the growth curves of *E. coli* and *S. aureus* were obtained. Generally, bacteria will have higher activity during the exponential growth phase, which consists of a viable state to test the antibacterial activity of the samples. Therefore, in each antibacterial experiment, the suspension was cultured to this phase. After the high−activity bacteria were obtained, the suspension was continuously diluted with PBS buffer until the colony concentration was 107–108 colony−forming units (CFU)/mL.

### 2.6. Antimicrobial Ability Test

#### 2.6.1. Antibacterial Ability Test

The antibacterial abilities of CS and C−CGAS were investigated by the colony counting method according to the ASTM E−2149−2013 standard, with slight modifications. *E. coli* and *S. aureus* were used as the model bacteria. First, 0.1 g each of CS and C−CGAS was

mixed with the diluted bacterial suspension, and the mixture was shaken in an incubator shaker at 37 °C and 200 r for 2 h. Afterward, 100 μL of the above suspension was pipetted onto a solid LB medium and cultured overnight at 37 °C. Finally, the number of colonies on each plate was counted. The diluted suspension without any samples was set as the control.

The antifungal method was similar to the antibacterial activity test. First, *Rhizopus stolonifera* was cultured on solid integrated potato medium at 30 °C for 1 week. Then, the spores were washed down, and approximately 1000 spores were added to 1 mL of 2% sucrose solution and 0.1 g of CS and C−CGAS was mixed with the spore suspensions. After culturing for 16 h at 30 °C, the mycelial growth in the suspension was observed by optical microscopy (BX51TF, Olympus Corporation, Tokyo, Japan).

### 2.6.2. SEM Observations of the Bacteria

The morphology changes in the bacteria before and after the antibacterial activity test were observed by SEM (scanning electron microscopy). The bacteria were washed three times with sterile water and fixed in a 10% PBS/glutaraldehyde mixture overnight. Then the bacteria were washed and dehydrated with a sequential ethanol/water mixture (30%, 50%, 70%, 80%, 90%, and 100%). After freeze−drying, the bacteria were observed by SEM.

### 2.6.3. Antifungal Ability Test

The antifungal ability method was similar to the antibacterial ability test. First, the *Rhizopus stolonifera* was cultured at 30 °C on a solid integrated potato medium for 1 week. Then, the spores were washed down, and approximately 1000 spores were added to 1 mL of 2% sucrose solution. Then, CS (0.1 g) and C−CGAS (0.1 g) were mixed with the spore suspensions, and after culturing for 16 h at 30 °C, the mycelial growth in the suspension was observed by optical microscopy (BX51TF, Olympus Corporation, Tokyo, Japan).

### 2.7. Water Purification Test

#### 2.7.1. Determination of Water Flux

The water flux ($J$) was calculated by the following equation:

$$J = \frac{V}{S \times T},$$

(1)

where $S$ (m$^2$) is the inner area of the tube, $V$ (L) is the volume of the permeated liquid, and $T$ (h) is the separation time. Each measurement was conducted three times, and the average value was obtained.

#### 2.7.2. Water Bacteria Disinfection Test

In this experiment, *E. coli* was chosen as the model bacterium to test the disinfection abilities of C−CGAS. The continuous circulation water purification device is shown in Figure S1, where C−CGAS was approximately 60 mm in height and 20 mm in diameter. The suspension was continuously diluted with PBS until the colony concentration was $10^4$–$10^5$ CFU/mL. C−CGAS was used to tightly fill the pipe, and 100 mL of the bacterial suspension was passed through the CS at a flux of 850 mL/h, using a constant flow pump. After passing through the CS, the disinfected bacterial suspension was pumped back into the original bacterial suspension and disinfected repeatedly. During disinfection, 100 μL of the disinfected suspension was pipetted onto the solid LB medium at 0.5, 1, 2, and 5 h. The plates were subsequently incubated at 37 °C overnight, and the number of bacterial colonies on each plate could be easily counted. The diluted suspension without purification was used as the control.

#### 2.7.3. Recycling Test of C−CGAS

Based upon the water bacteria disinfection test, the disinfection time was adjusted to 2 h, where C−CGAS was approximately 20 mm in height and 20 mm in diameter. The C−CGASs in this test were immersed in 75% ethanol for 15 min and washed with

deionized water completely to thoroughly disinfect them after each test cycle. Afterward, the disinfected suspension was pipetted onto the solid LB medium at 37 °C overnight, and the number of bacterial colonies was counted. This process was repeated nine times to determine the recyclable antibacterial activity. The diluted suspension without purification was used as the control.

### 2.8. Statistical Analysis

The experiment was arranged with a completely randomized design in triplicate, and SPSS 17 software was used for the analysis of variance (ANOVA). The differences between the mean values were analyzed by Duncan's test, and the differences were considered significant at $p < 0.05$. LSD was used to compare the difference significances between the mean values.

## 3. Results and Discussion

### 3.1. Morphological Characterization of CS

In this study, to combine the high water flux and mechanical properties of water disinfection materials, a hierarchical porous structure sponge was constructed. As shown in Figure 1a, the "long−chain" cellulose of AC was used to construct major pores as a skeleton, while the "short−chain" cellulose of MCC filled the gaps between the "long−chain" cellulose, constructing minor pores. The sponge structure was observed by SEM, as shown in Figure 1. Figure 1b,c shows that the AC sponge exhibited microscale pores and was constructed by microscale fibers 15.4 μm in diameter. For the MCC sponge, at low magnification, the MCC sponge exhibited a relatively dense network structure (Figure 1d). At high magnification, many nanopores accompanied by pore wall accumulation were observed in the MCC sponge (Figure 1e). Furthermore, the viscosity−averaged molecular weights of MCC and AC were determined, with values of $2.99 \times 10^2$ and $2.49 \times 10^3$, respectively (Figure S2). As shown in Figure 1f,g, the major pores constructed by AC and the minor pores constructed by MCC were simultaneously observed in CS, indicating that the hierarchical porous structure was successfully prepared. As a skeleton, AC could enhance the mechanical properties and ensure the structural stability of water purification materials [48], and MCC had a nano network with a high specific surface area, which increased the specific surface area and provided more active sites for the modification of CGA. Moreover, after modification with CGA, C−CGAS still maintained a hierarchical porous structure, as shown in Figure S3. Therefore, the hierarchical porous structure enabled C−CGAS to stably graft with the antibacterial agent while possessing mechanical stability, which contributed to the efficient disinfection of water.

Table 1 shows the specific surface areas (SSA) of the sponges. The AC sponge had the lowest SSA ($38.87 \text{ m}^2/\text{g}$), while the SSA of the MCC sponge was $67.12 \text{ m}^2/\text{g}$, which was higher than the AC sponge. Due to the hierarchical porous structure and MCC, which acted as a filler that filled the gaps between AC to construct minor pores, CS had the highest SSA ($131.10 \text{ m}^2/\text{g}$). After CGA modification, the SSA of C−CGAS was almost unchanged ($126.48 \text{ m}^2/\text{g}$).

**Table 1.** SSA of the AC sponge, MCC sponge, CS, and C−CGAS.

| Samples | Specific Surface Area ($\text{m}^2/\text{g}$) |
| --- | --- |
| AC sponge | 38.87 |
| MCC sponge | 67.12 |
| CS | 131.10 |
| C−CGAS | 126.48 |

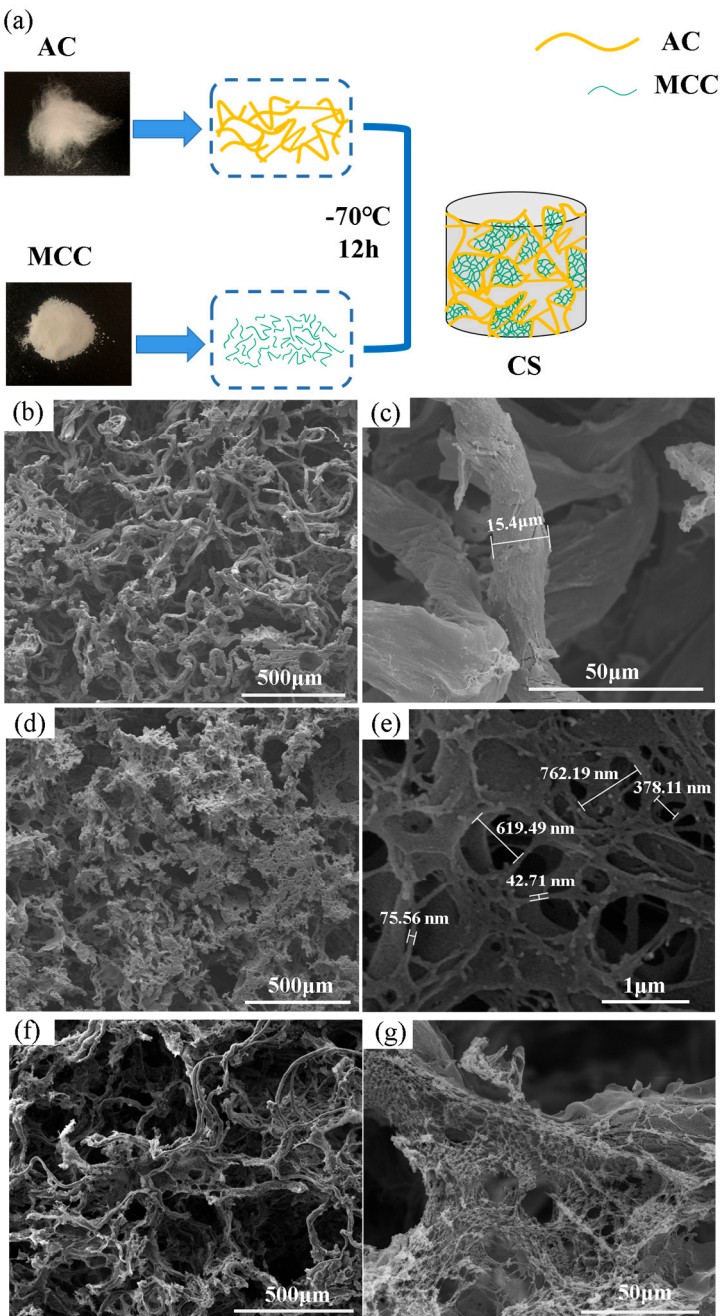

**Figure 1.** (**a**) The preparation of CS and SEM images of the (**b**,**c**) AC sponge, (**d**,**e**) MCC sponge, and (**f**,**g**) CS.

### 3.2. Characterization Analysis of CS and C−CGAS

Scheme 1 shows the modification of CGA, in which epichlorohydrin (EPI) acted as a crosslinker and covalently linked CS and CGA together through the hydroxyl groups, achieving the stable modification of CGA. To demonstrate the successful modification of CGA on CS, CS and C−CGAS were characterized. As shown in Figure 2a, an apparent color change was clearly observed after CS was successfully modified by CGA, where the color of CS changed from white to brown after modification. Figure 2b shows the FTIR spectra of CS, CGA, and C−CGAS. The spectra of CS and C−CGAS showed typical absorption bands of cellulose at 3447, 2920, and 1066 cm$^{-1}$, which were attributed to the −OH, −CH$_2$−, and C−O−C bonds, respectively [49]. The CGA showed typical phenolic characteristics, in which the peaks at 1688 cm$^{-1}$ were attributed to the vibrations of mixed

ester and carboxyl C=O. The 3383 cm$^{-1}$ peaks were due to −OH vibrations, while the carboxyl C−O−C stretching vibrations were found at 1285 and 1185 cm$^{-1}$, the carboxyl O−H bending vibrations were found at 603 cm$^{-1}$, and the benzene ring stretch vibrations were found at 1688, 1602, 1518, and 1442 cm$^{-1}$ [50]. In addition, for C−CGA, absorption bands at 3447, 2920, and 1066 cm$^{-1}$ were observed, which were attributed to cellulose, and CGA bands at 1630, 1610, 1511, and 1454 cm$^{-1}$ were observed, indicating that CGA was successfully modified on the CS.

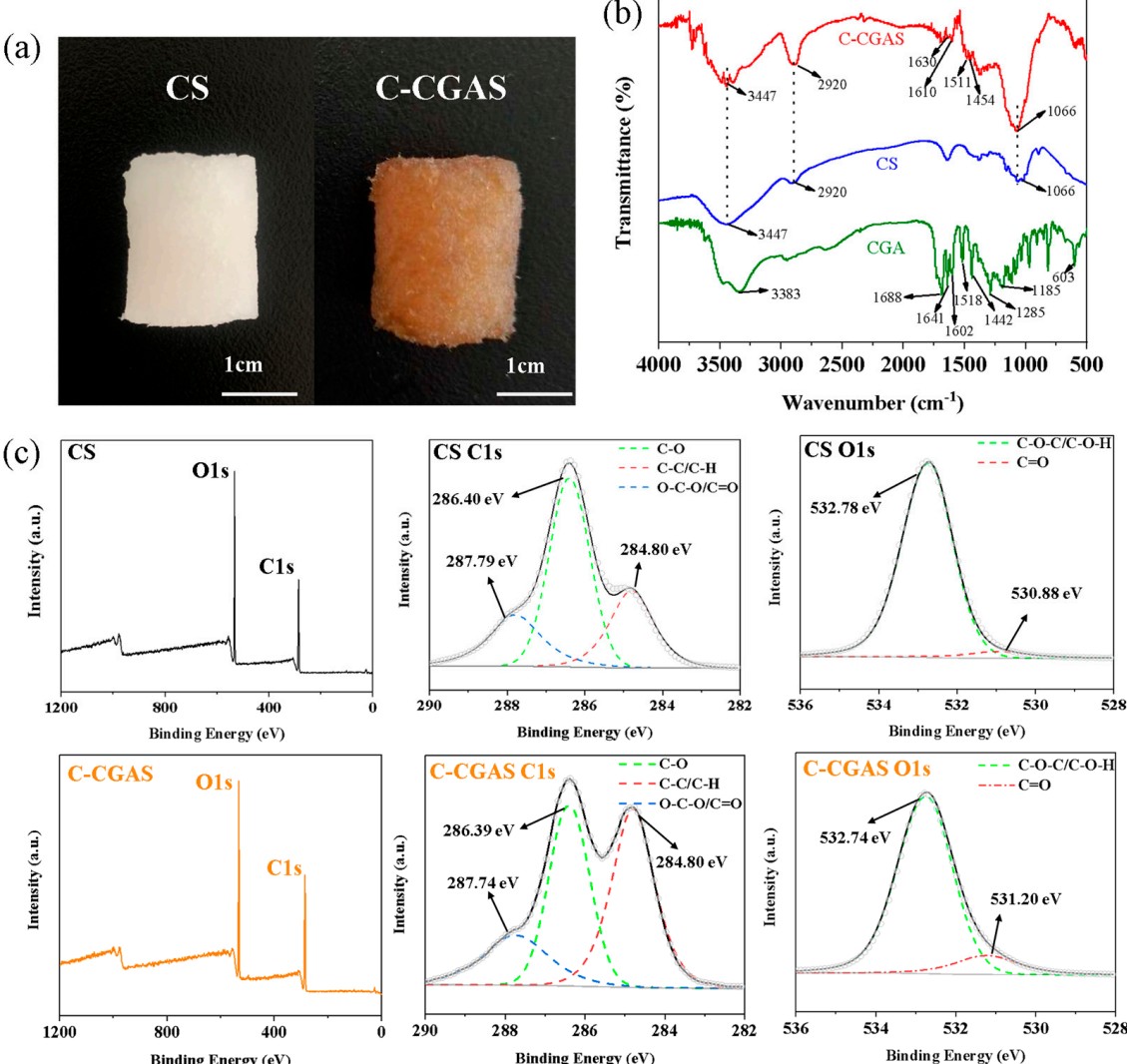

**Figure 2.** (**a**) Digital images of CS and C−CGAS; (**b**) FTIR spectra of CS, CGA, and C−CGAS; the XPS survey spectra showing C1s and O1s and XPS spectra of (**c**) CS and C−CGAS.

Figure 2c shows the XPS results of CS and C−CGA. In the C1s spectra of CS, the peaks at 287.79, 286.40, and 284.80 eV corresponded to O−C−O/C=O, C−O, and C−C/C−H, respectively [51]. The O1s spectra of CS showed two peaks at 532.78 and 530.88 eV, which were attributed to C−O−C/C−O−H and C=O, respectively [52]. Similarly, these peaks were also observed in the C−CGAS spectra, but the content of C=O and C−C/C−H increased, which was due to carboxyl and carbon from the covalently linked chlorogenic acid. The decrease in C−O−C/C−O−H was due to hydroxyl consumption when CS reacted with EPI. These results further demonstrated that CS was successfully modified by CGA, and the contents changed greatly, as shown in Table S1.

### 3.3. Hydrophilicity and Physical Properties

The hydrophilicity of the filter will directly affect its water disinfection effect; therefore, the hydrophilicity of the sponges was tested according to the dynamic water contact angle (Figure 3a). The water droplet permeated CS and C−CGAS in only 0.213 and 0.261 s, respectively, suggesting acceptable hydrophilicity, which was attributed to a large number of hydroxyl groups present in cellulose. In addition, to test the diffusion performance of water in the materials, 500 μL of Congo red (0.05 g/L) was injected into CS and C−CGAS. As shown in Figure 3b, Congo red diffused deeply into both sponges, indicating that both CS and C−CGAS had adequate water diffusivity. Sufficient hydrophilicity and water diffusivity allowed the antimicrobial material to effectively contact the bacteria in the water, and the disinfection ability of the filter was greatly improved.

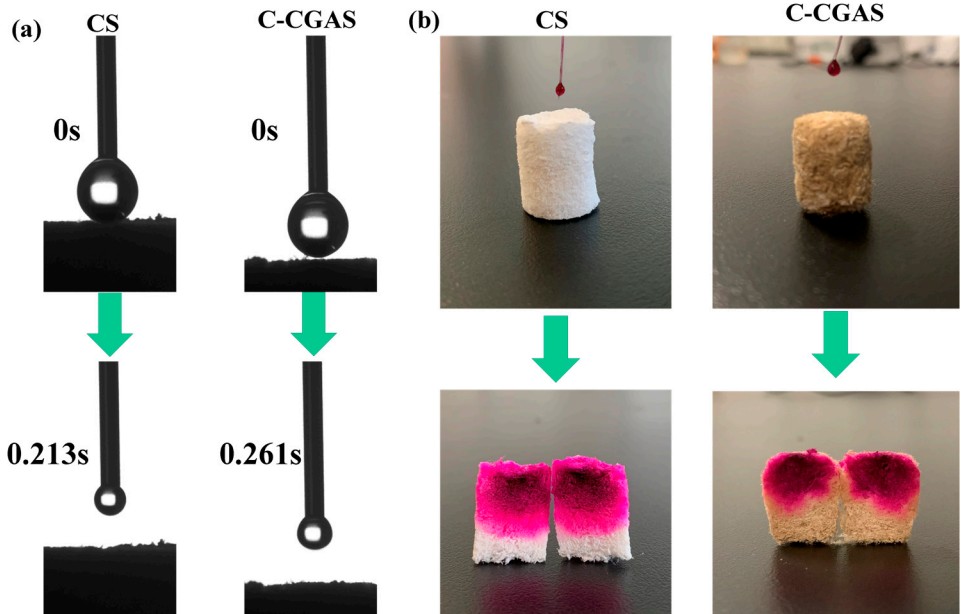

**Figure 3.** (**a**) The dynamic water contact angles of CS and C−CGAS; (**b**) digital images of the dye diffused in the freeze−dried CS and C−CGAS.

In addition, the physical properties of CS and C−CGAS were tested. As shown in Table S2, the bulk density and porosity of CS and C−CGAS were similar, with values of 46.54 and 47.88 mg/cm$^3$, and 97.14% and 97.09%, respectively. Furthermore, the moisture content and water absorption values of CS and C−CGAS were 77.81% and 78.53%, and 20.73 and 20.16 g/g, respectively. In summary, CS and C−CGAS exhibited similar physical properties, indicating that CS modified by CGA still maintained its original characteristics, such as high porosity and water absorption, which were beneficial to the hydrophilicity and disinfection efficiency of the filter.

### 3.4. Mechanical Property Tests

As a water disinfection material, mechanical strength is important for stability and service life; therefore, the mechanical properties of the sponges were tested (Figure 4a). Figure 4b shows the compressive stress–strain curves of the sponges, and Figure 4c shows the compressive modulus values of the different sponges. All of the sponges displayed three stages characteristic of a foam−like structure: a linear elastic region (below strain of 5%), a plastic region where the curve was relatively flat (strain of more than 5% but less than 55%), and a densification region with the acceleration of stress growth (overstrain of 55%).

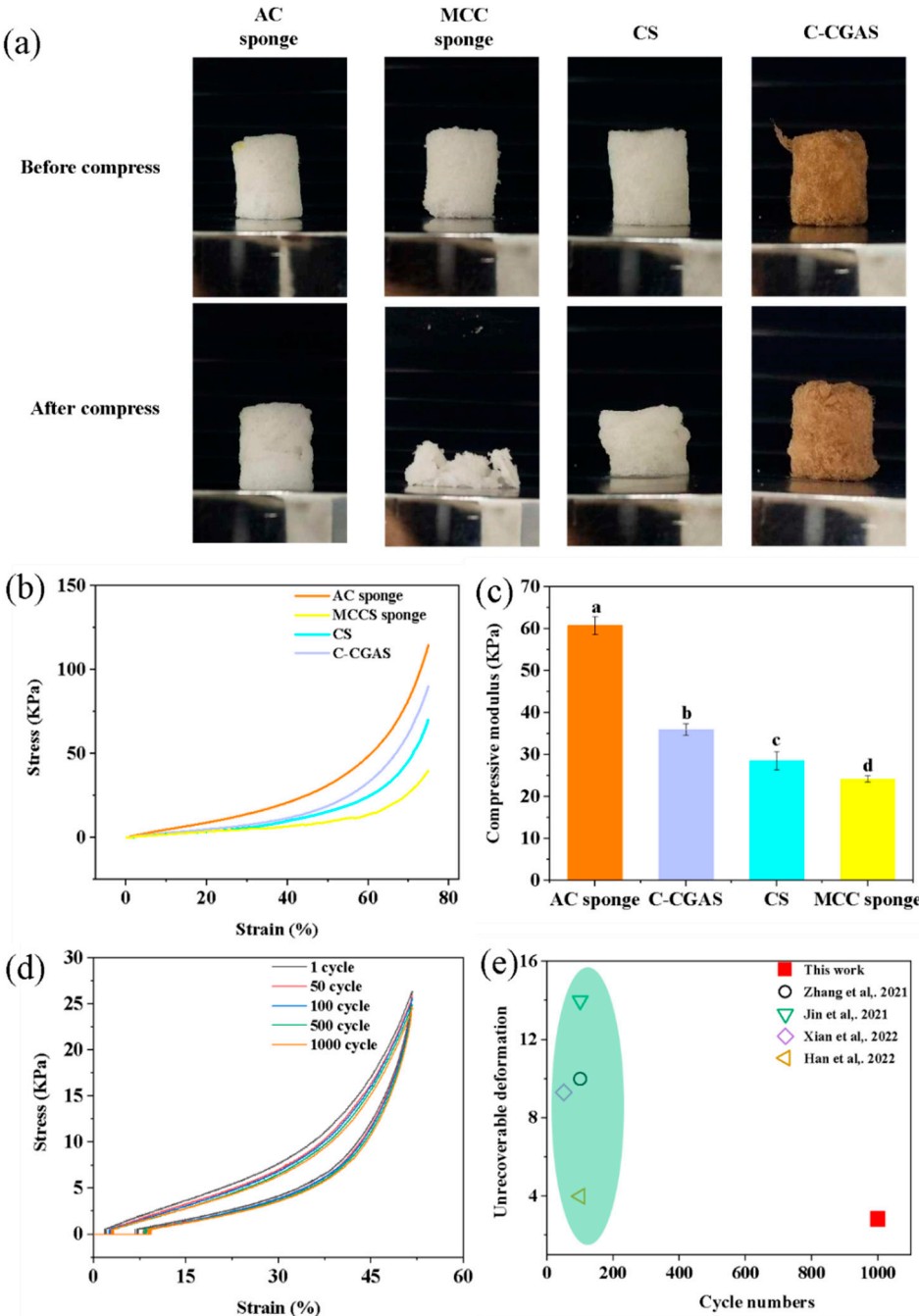

**Figure 4.** (**a**) Images of CS and C−CGAS before compression and after compression; (**b**) compressive stress–strain curves of the MCC sponge, AC sponge, CS, and C−CGAS; (**c**) the compressive modulus and (**d**) cyclic compression test of the C−CGAS after 1000 cycles; and (**e**) comparison of cyclic compression stability with other reports.

As shown in Figure 4c, the compressive modulus of the AC sponge was 60.59 kPa, with no breakage after compression (Figure 4a), showing the robustness of the AC sponge, which was attributed to the construction of the AC sponge made of continuous strong crude fibers [53]. The MCC sponge was softer than the AC sponge, where the compressive modulus of the MCC sponge was 24.88 kPa. However, the sponge was not elastic after compression, and serious damage was observed, which was attributed to the pore walls forming a dense sheet layer, and the porous structure was lost (Figure 4a,c). Figure 4c shows that the compressive modulus of C−CGAS (35.94 kPa) was slightly higher than CS (28.53 kPa). We also found that these two sponges almost returned to their original shape,

even when the compressive strain was as high as 75%, suggesting the suitable elasticity and stability of CS and C−CGAS (Figure 4a). Although the AC sponge had higher stability than CS, due to the small specific surface area of the sponge, the modified content of the antibacterial agents was not sufficient for use as water disinfection materials; therefore, the AC sponge was not suitable for water disinfection. In addition, the compressive modulus of C−CGAS was slightly higher than CS, which was attributed to the epichlorohydrin crosslinks of some hydroxyl groups in cellulose, which improved the mechanical strength of the sponge.

To study the mechanical stability of C−CGAS, cyclic compression experiments were carried out. Figure 4d shows the cyclic compressive stress–strain curves of C−CGAS, where after 1000 compression cycles, the ultimate stress was maintained at 24.65 kPa, which was 93.37% of the original stress (26.40 kPa). In addition, C−CGAS produced only 2.84% unrecoverable deformation after 1000 compression cycles, which was much smaller than other reported sponge materials, demonstrating adequate cycling mechanical stability (Figure 4e) [54–57]. Therefore, benefiting from the mechanical properties and cyclic compression stability, breakage of the sponges was effectively prevented during the disinfection process, extending the service life of water disinfection.

### 3.5. Antimicrobial Ability Test

### 3.5.1. Antibacterial Ability Test

Microbial contamination in water is a severe imperilment to human health; therefore, effectively killing bacteria is an important indicator for water disinfection materials. The antibacterial abilities of C−CGAS were evaluated by dynamic contact antibacterial testing. *E. coli* was used as the model Gram−negative bacterium, and *S. aureus* was used as the model Gram−positive bacterium. Figure 5a presents the digital images of the coated agar plates after incubation. We observed that a large number of colonies grew on the agar plates of the control group, whereas the number of colonies on the agar plates of the CS group was similar to the control group, which proved that cellulose had no antibacterial ability. However, no colonies were observed on the agar plates of the C−CGAS group, proving that C−CGA had adequate antibacterial properties. Bacterial reduction is shown in Figure 5b, which was compared to the CS group, indicating that bacterial reduction in the C−CGAS group was significantly higher.

To further investigate the changes in cell morphology before and after the antibacterial test, the bacteria were observed by SEM. As shown in Figure 5c, the normal bacteria were morphologically intact. However, the bacteria in the C−CGAS group showed a loss of membrane integrity, cell distortion, and even dissolution into clumps, which demonstrated the bactericidal ability of C−CGAS against *E. coli* and *S. aureus*. The antibacterial properties of CGA were because CGA could bind to the outer membrane of the bacteria, disrupting the membrane, exhausting the intracellular potential, and releasing cytoplasm macromolecules, which led to bacterial death [43].

### 3.5.2. Antifungal Ability Test

In addition to the antibacterial ability, the antifungal ability has been shown to be particularly important for water disinfection materials. Therefore, the antifungal abilities of CS and C−CGAS were determined. In this test, *Rhizopus stolonifera* was used as the model fungus. As shown in Figure 5d, hyphae growth in the control was clearly observed, as indicated by the red arrows, proving that spore germination was not inhibited. The same phenomenon of hyphal growth was also observed in the CS group, indicating that CS had no antifungal properties. However, no hyphae growth was observed in the C−CGAS group due to the inhibition of spore germination, proving that C−CGAS had adequate antifungal properties. The antibacterial properties of CGA could be attributed to the fast induction of membrane permeabilization in the fungal spores by CGA [58].

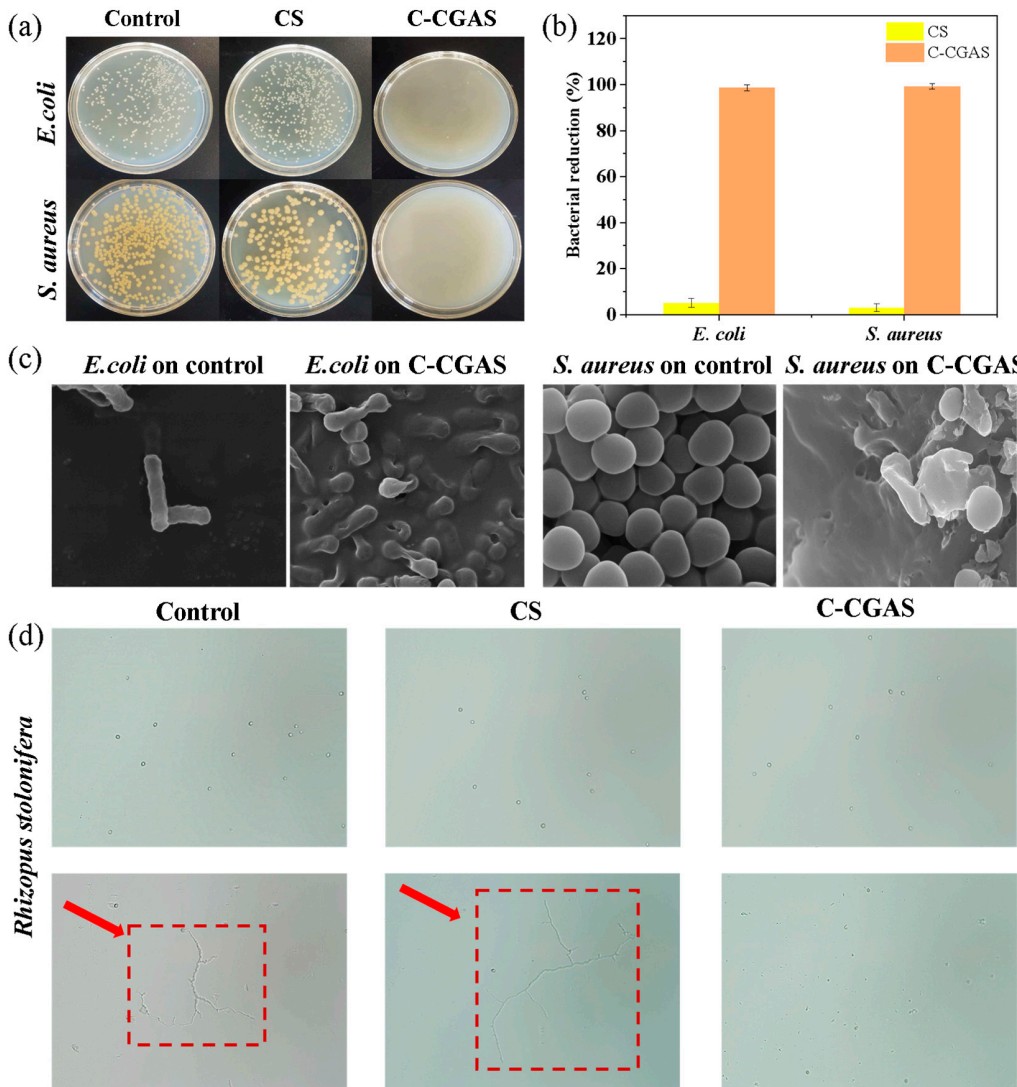

**Figure 5.** (**a**) Images of the agar plates of the bacterial suspensions cultured with CS and C−CGAS; (**b**) bacterial reduction of suspensions cultured with CS and C−CGAS; and (**c**) SEM images of the bacteria and (**d**) fungi in contact with CS and C−CGAS.

### 3.6. Water Disinfection Test of C−CGAS

As one of the most abundant beneficial polyphenols in plants, the antibacterial properties of CGA have been proven [59,60]. Due to the biological safety of CGA, it is widely used in health products and food additives [44,61,62]. In addition, CGA can be bound to CS through covalent modification. In this work, after modification, no CGA was detected in the eluent even after vigorous stirring in 100% ethanol and 50% ethanol for 3 days, proving the stability of the modification (Figure S4). Due to the antibacterial properties and stable modification of CA, C−CGAS had a stable antibacterial ability and could realize high−efficiency water disinfection.

Another important factor for the high efficiency of water disinfection materials is high water flux. In the water disinfection test in this study, each sponge was cylindrical, approximately 20 mm in diameter and 20 mm in height, and the number of sponges in use ranged from 1 to 3. Figure 6a shows the water flux analysis of CS and C−CGAS at different layers, where C−CGA and CS with the same number of layers had similar water fluxes, indicating that the modification of CGA did not affect the porous structure of the sponges. In addition, with an increase in sponge number, although the water flux of CS and C−CGAS decreased, three−layered CS and C−CGAS still had water flux values up

to $0.93 \times 10^5$ and $1.0 \times 10^5$ L·m$^{-2}$·h$^{-1}$·bar$^{-1}$, respectively. Even though the water flow pushed through the C−CGAS at a high flow rate of 30 mL/min, only a low pressure of 0.66 kPa was generated (Figure S5), which was attributed to the hydrophilicity and high porosity of the sponge.

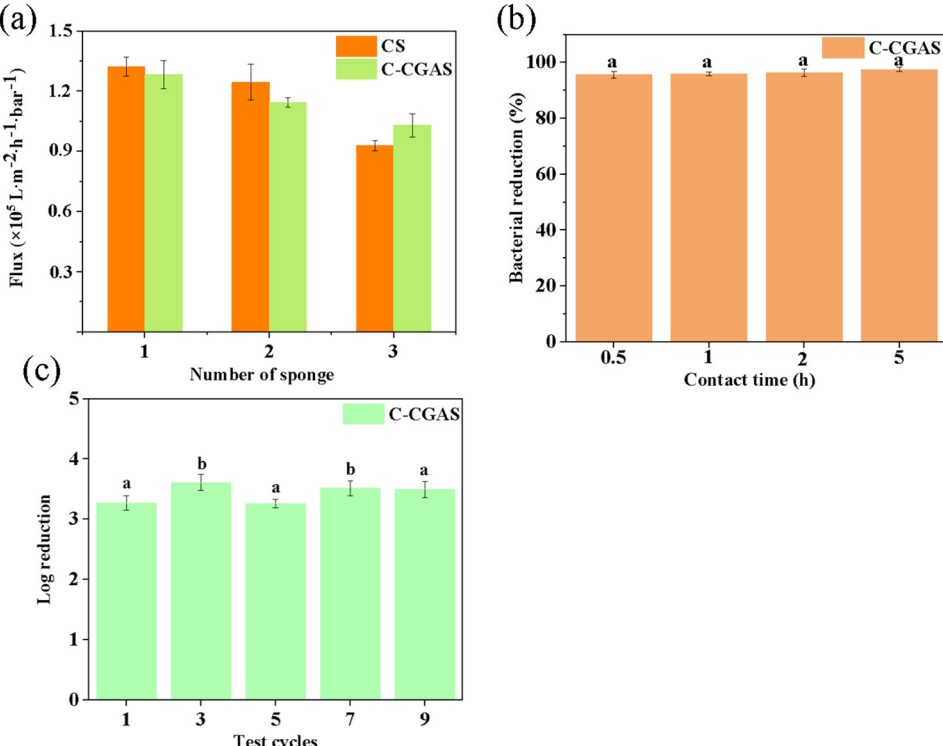

**Figure 6.** (**a**) Histograms of CS and C−CGA water flux; (**b**) bacterial reduction of suspensions disinfected with CS and C−CGAS; and (**c**) cyclic antibacterial ability of C−CGAS. Letters a, b indicate significant differences ($p < 0.05$) using Duncan's test.

To gain insight into the water flux of the sponge, the water flux of the sponge in this research could be explained by the Hagen–Poiseuille equation [50]:

$$J = \frac{\pi \eta r_p{}^2 \Delta P}{8 \mu L} \tag{2}$$

where the flux $J$ is related to the porosity $\eta$, the pore radius $r_p$, the applied pressure $\Delta P$, the liquid viscosity $\mu$, and the sponge thickness $L$. In our experiments, $\Delta P$ was fixed at 1 bar, and the viscosity of water was unchanged. Compared to the other water disinfection membranes, the hydrophilicity and unique structure with high porosity, as well as a large pore radius, provided the sponge with high water flux.

To further study the water disinfection effect of C−CGAS, *E. coli* was chosen as the model bacterium and was added to deionized water to simulate sewage. Then, the water containing *E. coli* was passed through a continuous circulation water purification device to test the water bacteria disinfection abilities of CS and C−CGAS. As shown in Figure 6b, after 0.5 h of disinfection, bacterial reduction was over 95.63%, and bacterial reduction reached 97.05% after 5 h, indicating that C−CGAS had a rapid water disinfection ability.

Stable disinfection ability and reusability are considered important for the cyclic service life of water disinfection materials. Therefore, cyclic antibacterial testing was carried out on C−CGAS. As shown in Figure 6c, after the sponges disinfected the water containing bacteria for 2 h over each test cycle, the log reduction of C−CGAS was determined, where 1 log reduction resulted in 10−fold (one decimal place) or 90% reduction in CFU, and a higher log reduction indicated a better water antibacterial treatment ability. The log

reduction of cellulose–chlorogenic acid sponge (C−CGAS) was in the range of 3.27–3.51 in the nine cycles, indicating that the antibacterial ability was stable (Figure S6). So far, silver nanoparticles and chlorine have been widely used in water disinfection materials. Compared to the silver nanoparticle water disinfection materials (log reduction less than 3.00) [63–65], C−CGAS had a higher log reduction. Although chlorine−modified water disinfection materials showed a higher log reduction (higher than 5.00) than C−CGAS, the chlorine that remains in the water may have serious impacts on human health [66–68]. Therefore, C−CGAS could be used as a reusable, efficient, and environmentally friendly water disinfection material.

## 4. Conclusions

A CGA−modified hierarchical cellular structure CS was successfully prepared for water disinfection. The hierarchical cellular structure was constructed by AC and MCC, in which AC acted as the skeleton and was used to construct major pores, while MCC acted as the filler, which filled in the gaps between the "long−chain" cellulose to construct minor pores. Benefiting from the hierarchical cellular structure, C−CGAS showed no breakage at a high strain rate of 75%. Even after 1000 compression cycles, only 2.84% of the unrecoverable strain was produced. Furthermore, as an ideal water disinfection material, C−CGAS exhibited high water flux ($1.28 \times 10^5$ L·m$^{-2}$·h$^{-1}$·bar$^{-1}$) and adequate antibacterial and antifungal abilities. In the water disinfection test, the bacterial reduction was over 98% within 0.5 h. Because of the stable covalent modification of CGA on the sponges, after nine consecutive use cycles, the antibacterial properties were almost unchanged (log reduction between 3.27 and 3.51). This work fabricated a biosafe and sustainable high−efficiency antibacterial material with sufficient mechanical strength and high water flux, providing a new strategy in the field of water disinfection material preparation.

**Supplementary Materials:** The following supporting information can be downloaded at: https://www.mdpi.com/article/10.3390/su15010773/s1, Figure S1: The digital image of the device used in water bacteria disinfection test and cyclic bactericidal tests; Figure S2: $\ln\eta_r/c$ vs. c of (a) absorbent cotton and (b) microcrystalline cellulose in CED at 25 °C ([$\eta$]$_{AC}$ = 777.61 cm$^3$/g, [$\eta$]$_{MCC}$ = 151.77 cm$^3$/g); Figure S3: SEM images of the C−CGAS at (a) low magnification and (b) high magnification; Figure S4: HPLC chromatograms of the extractants after immersing C−CGAS for 3 days and ethanol standard solution (STD); Figure S5: Digital image of C−CGAS water flux stress testing device; Figure S6: Digital images of agar plates of bacterial suspensions in cyclic antibacterial ability test; Table S1: Summary of XPS spectral parameters of CS and C−CGAS; Table S2: The density, porosity, moisture content, and water absorption of the cellulose sponges. Ref. [69] has cited in Supplementary Materials file.

**Author Contributions:** Conceptualization, D.-Y.Z. and T.C.; methodology, R.-Z.H., E.-J.L. and J.-X.H.; investigation, E.-J.L. and J.-X.H.; formal analysis, E.-J.L. and X.-H.Y.; resources, T.C. and D.-Y.Z.; writing, E.-J.L.; software, E.-J.L. and W.-G.Z. All authors have read and agreed to the published version of the manuscript.

**Funding:** This work was supported by the Six Talent Peaks Project in Jiangsu Province (SWYY-155), the Postdoctoral Research Fund Project of Jiangsu Province (2021K253B), the Young Scholars Program of Jiangsu University of Science and Technology, the earmarked fund for CARS-18, the National Key R&D Program of China, Key Projects of International Scientific and Technological Innovation Cooperation (2021YFE0111100), the Guangxi Innovation-Driven Development Project (AA19182012-2), and the Zhenjiang Science and Technology Support Project (GJ2021015).

**Institutional Review Board Statement:** Not applicable.

**Informed Consent Statement:** Not applicable.

**Data Availability Statement:** Not applicable.

**Acknowledgments:** The authors would like to thank College of Biotechnology and Sericultural Research Institute and Jiangsu University of Science and Technology for device and technical support.

**Conflicts of Interest:** The authors declare no conflict of interest.

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
