# Peer review of "A Hierarchical Porous Cellulose Sponge Modified with Chlorogenic Acid as a Antibacterial Material for Water Disinfection"

_sustainability, doi:10.3390/su15010773_

Round 1
Reviewer 1 Report
In this paper, cellulose-based materials were constructed using covalent modification and their porous structure were characterized using different techniques (SEM, FTIR, XPS, contact angles…). Cellular sponge materials (CGA) showed high mechanical resistance with good shape recovery properties and high water flux. Such sponge materials were successfully tested for water disinfection.
This paper is well written with a good quality of data presentation. Cellulose sponge is a promising materials for water disinfection. The paper can published.
Reviewer 2 Report
This manuscript describes the elaboration of a modified porous cellulose sponge for water disinfection. Cellulose is modified with epichlorhydrin to graft chlorogenic acid in cellulose units. Several experimental results show the efficient modification of cellulose sponge. According to the authors, C-CGAS sponge present good mechanical properties with a good stability (cyclic compression experiments). C-CGAS sponge shows antibacterial and anti-fungal activities. The antibacterial activity was tested for several cycles.
overall comment :
This paper fits very well to the scope of the journal. The propose approach seems to be novel and has a potential impact on the community. A comparison with other water purification materials would have been interesting.
minor comments :
- l50 : cationic polymers is repeated.
- l50 references for each example (cationic polymers, antibacterial peptides, metal nanoparticles ...) should be cited.
- references should be chosen specifically for water purification materials (ref 14, 15)
- It can be interesting to cite some previous works about the introduction of chlorogenic acid on polysaccharides in general.
- Scheme 1 should be reviewed
- It will be interesting to add the degree of substitution of chlorogenic acid (XPS analysis or elementary analysis may help)
Reviewer 3 Report
Thank you for the opportunity to review the manuscript entitled ”A hierarchical porous cellulose sponge modified with chlorogenic acid for safe and efficient water disinfection” aimed at development of water disinfection materials based on cellulose-based resources.
Although the study presents interesting data which seem to have been carefully acquired, it is regrettable that the manuscript is written in a wrong manner rather than organized around the cellulose as a biopolymer.
The main drawback for the research to be relevant is confusion of the terms used. It would be recommended that the authors assess the potential of their data and rewrite the manuscript having in mind that cellulose is a polymer, while cellulose sponges is a material. Then, saying that ”The frozen mixture was immersed in hot water until the Na2SO4, NaOH, and urea were completely REMOVED, and CS were obtained.” denote the fact that the work is of poor scientific quality such that it is clearly not suitable for publication in a top-tier scientific journal.
In brief, the data presented in this manuscript are interesting, the paper is well-written, well-structured, and richly illustrated, and some parts of this study represent an interesting contribution to the field. However, I do not feel that this research as it was submitted throws new light that deserves publication in a top-tier scientific journal devoted to advanced studies related to sustainability and sustainable development.
Reviewer 4 Report
The work is devoted to the preparation of novel hierarchical porous cellulose sponge as material for water disinfection. The authors propose to combine two types of cellulose differing in the degree of polymerization to create a hierarchical porous structure. The field of research lies in the scope of the journal. The authors have achieved the result, indeed, the cellulose-based material which was designed and prepared in the work has good operational properties for using it as a filter for water purification. Nevertheless, I have questions about the design of the article and the description of the results.
The authors did not cover the issue of creating cellulose-based filter materials well enough. There is plenty of works devoted to obtaining cross-linked cellulose sponges as filter materials. For example, cellulose cross-linked with EPI: Udoetok, Inimfon A., et al. "Adsorption properties of cross-linked cellulose-epichlorohydrin polymers in aqueous solution." Carbohydrate polymers 136 (2016): 329-340.)
Also, there is no well-written background of designing and preparing similar materials with complex porosity. In my opinion the intro must be substantially revised in such way that reader understands what has already been done in this area and what needs to be improved. English also should be improved.
I have some key issues to the authors that definitely need to be clarified:
1) In the methodology, the authors indicate microcrystalline cellulose and absorbent cotton as two polymer components for forming the hierarchical structure of the porous sponge.
Firstly, microcrystalline cellulose varies greatly from manufacturer to manufacturer, so specify its output data (specification data: degree of polymerization, particle size, hemicellulose content)
Secondly, it is not entirely clear what absorbent cotton is. Is it a trade name? Does this product have a specification, what are the characteristics of this cellulose (degree of polymerization, particle size, hemicellulose content)?
2) Please explain what you were guided by when choosing the material (MCC and absorbent cotton) for the sponge frame?
It does not seem quite clear for me why there is a need to create a complex hierarchical sponge structure from a material that already has good absorbing properties. Well, I suppose absorbent cotton possess good absorbing capacity. Have you compared your material with absorbent cotton? In the article, give a comprehensive explanation of why it is necessary to destroy (dissolve) the material (absorbent cotton) with good absorbing properties.
In my opinion authors should make additional experiments to compare the absorbing properties of composite sponge (based on two celluloses) and separate components.
The manuscript contains lots of inaccuracies some of which (not all of them) are listed below:
1. “Figure 6. (a) Digital images of CS and C-CGA water flux; (b) bacterial reduction of suspensions disinfected with CS and C-CGAS; and (c) cyclic antibacterial ability of C-CGAS.”
Please be careful, it is not digital images. Digital images are photographs. Figure 6 contains histograms.
2. “Figure 1. (a) The preparation of CS and SEM images of the (b, c) AC sponge, (d, e) MCC sponge, 246 and (f, g) CS. (h) SSA of the AC sponge, MCC sponge, CS, and C-CGAS.”
In my opinion it will be better SSA data to present as a table not the histogram.
3. In my opinion the way to depict the scheme of interaction of EPI and cellulose chains is not the best. It is better to depict the product as cellulose chain as long chains parallel each other with cross-links with fragments of EPI. And also, please explain what happened with cross-linked cellulose after reaction with chlorogenic acid? Absolutely unclear from the scheme.
4. “As one of the most abundant beneficial polyphenols in plants, the antibacterial prop-378 erties of CGA have been proven [33, 47]. Due to the good biological safety of CGA, it is 379 widely used in health products and food additives [34, 48, 49].”
In section 3.6 authors give the reader information about chlorogenic acid. The references are listed. Authors should avoid inserts from the literature review when discussing the results.
Round 2
Reviewer 3 Report
Dear authors, sice i only received 3 days to reply to your research, on short my main complain is, as stated before, thhat the work is of poor scientific quality such that it is clearly not suitable for publication in a top-tier scientific journal.
Mainly:
- the articles is around material science, not chemical science;
- the article is not about cellulose, because cellulose is a polymer; you confuse the terms, you use a material that translated from one langiuage to another makes confusion on terms - see other terms like pulp, fibers, fibres to use...
- Adsorbent cotton is not a polymer, is a material - it is impossible to make a chemical reaction and to say that ”chair + cl2 = 2chair-cl”;
- you should rename the title of the article to be correlated to the content...
Best wishes!
